# A Rare Case of Endophthalmitis with *Rhizobium radiobacter*, Soon after a Resolved Keratitis: Case Report

**DOI:** 10.3390/antibiotics11070905

**Published:** 2022-07-06

**Authors:** Andrei Theodor Balasoiu, Ovidiu Mircea Zlatian, Alice Elena Ghenea, Livia Davidescu, Alina Lungu, Andreea Loredana Golli, Anca-Loredana Udriștoiu, Maria Balasoiu

**Affiliations:** 1Department of Ophthalmology, University of Medicine and Pharmacy of Craiova, 200349 Craiova, Romania; andrei_theo@yahoo.com; 2Department of Bacteriology-Virology-Parasitology, University of Medicine and Pharmacy of Craiova, 200349 Craiova, Romania; balasoiu.maria@yahoo.com; 3Department of Ophthalmology, County Clinical Emergency Hospital of Craiova, 200642 Craiova, Romania; liviadavidescu@yahoo.com; 4Department of Infectious diseases, County Clinical Emergency Hospital of Craiova, 200642 Craiova, Romania; marinaalina@gmail.com; 5Department of Epidemiology, University of Medicine and Pharmacy of Craiova, 200349 Craiova, Romania; andreea_golli@yahoo.com; 6Faculty of Automation, Computers and Electronics, University of Craiova, 200776 Craiova, Romania; anca.udristoiu@edu.ucv.ro

**Keywords:** *Rhizobium radiobacter*, endophthalmitis, cataract surgery, corneal trauma

## Abstract

Background: *Rhizobium* (*Agrobacterium*) species are plant aerobic bacteria, which in some cases can produce endophthalmitis in humans after corneal trauma. Case presentation: A 42-year-old female patient presented in the Emergency Department of the Emergency County Hospital of Craiova, Romania, reporting pain, epiphora, and blurry vision in her right eye for about five days. This initial infectious keratitis episode was successfully resolved, but after 20 days she presented again after trauma with a leaf with corneal abscess. In the conjunctival secretion, *R. radiobacter* was identified. Despite antibiotherapy, the patient’s state did not improve, and ultimately the eye was eviscerated. Methods: A search was performed in the ProQuest, PubMed, and ScienceDirect databases for the terms Agrobacterium, Rhizobium, radiobacter, and eye. We eliminated non-human studies, editorials and commentaries, and non-relevant content, and excluded the duplicates. Results: In total, 138 studies were initially obtained, and then we selected 26 studies for retrieval. After the selection process, we ended up including 17 studies in our analysis. Most studies reported *R. radiobacter* endophthalmitis after ocular surgical procedures or outdoor activities that involve exposure to soil. Conclusion: *R. radiobacter* is a rare cause of endophthalmitis after eye trauma that generally responds well to usual antibiotherapy, but occasionally can evolve to severe, leading to the loss of the eye.

## 1. Introduction

Endophthalmitis is a leading cause of eye loss worldwide, despite the prophylactic antibiotherapy used for eye trauma, whether accidental or surgical [1,2,3,4]. Table 1 summarizes the main infectious agents involved in endophthalmitis, which is caused by different mechanisms, including trauma, eye surgery, contact lenses, eye injections or other related procedures, or hematological inoculation.

A rare cause of endophthalmitis is infection with *Rhizobium (Agrobacterium) radiobacter*, which is a plant aerobic bacterium belonging to the Gram-negative bacilli, which are common in the environment—especially in soil [21]—and cause plant diseases such as gall tumors and hairy root disease. Several cases of *Rhizobium radiobacter* endophthalmitis have been reported, usually after cataract surgery, but have only rarely been reported to produce human infections, most often in immunocompromised patients [22,23,24,25]. The main risk factors are non-adherence to aseptic techniques [26,27], paracentesis of the anterior chamber, and eyelid manipulation [2,28].

Here, we describe a case of bacterial keratitis with severe evolution.

## 2. Case Report 

A 42-year-old female patient presented in the Emergency Department of the Emergency County Hospital of Craiova, Romania, on 4 April 2022, reporting pain, epiphora, and blurry vision in her right eye for about five days. Anamnesis revealed a history of chemically treated soil contact with her right eye about seven days before the presentation, as she lives in the countryside and works in agriculture. The patient had not been immunocompromised or treated with steroids. Uncorrected visual acuity was 0.25 in her right eye and 1 in her left eye; the best-corrected visual acuity (BCVA) in her right eye was 0.25 (normal visual acuity is 1). Intraocular pressure (IOP) was 17 mmHg in her right eye and 15 mmHg in her left eye (the normal range is between 10 and 21 mmHg). Slit-lamp examination in the right eye revealed conjunctival hyperemia, a 4 mm white–yellow central corneal ulceration with infiltrated margins that retained the methylene blue stain, Descemet folds at the lesion and endothelial edema, a normal-depth anterior chamber, less than five cells in the aqueous, miotic, and reflexive pupil (Figure 1). Due to corneal haze, fundus examination was difficult to perform in the right eye, but it was normal in the left eye. Corneal sensitivity was tested in the right eye, and the result was hyperesthesia. A clinical diagnosis of infectious keratitis of the left eye was established. A SARS-CoV-2 rapid antigen test was performed, and as it was negative the patient was admitted to hospital in the Ophthalmology Clinic.

Conjunctival secretion and corneal scraping samples were collected and sent to the hospital’s laboratory for microbiology diagnosis and antibiogram; standard blood tests were also performed. Intravenous empirical broad-spectrum antibiotherapy was introduced immediately after sampling (third-generation cephalosporin/beta-lactamase inhibitor—cefoperazone/sulbactam), doubled by local antibiotherapy (topical moxifloxacin). Lubricants, mydriatics, topical NSAIDs, and eye patching were also prescribed. 

Microbiology results were negative for bacteria and fungi; the rest of the tests were also normal. Fortunately, the evolution was favorable: the ulceration’s size started to decrease daily, the Descemet folds became thinner, and visual acuity was also improving, so on 10 April 2022, the patient was discharged from hospital. Clinical examination of the right eye that day revealed that the uncorrected visual acuity was 0.9, IOP was 16 mmHg, the slit-lamp examination showed minimal conjunctival hyperemia, paracentral corneal 3 mm leucoma, and a quiet anterior chamber, and the fundus was also normal. The recommendation was topical treatment with lubricants, moxifloxacin, and NSAIDs, and ambulatory appointment after one week. 

The patient presented ambulatory, one week after hospital discharge, in the Ophthalmology Clinic, where the slit-lamp examination revealed a quiet right eye, no conjunctival hyperemia, paracentral corneal 3 mm leucoma, normal aspect of the fundus, and 0.9 visual acuity. Topical antibiotics and NSAIDs were interrupted after this examination, but the eye lubricants were kept and the patient returned to work. 

On 29 April 2022, the patient presented again in the Emergency Department of the Emergency County Hospital of Craiova, Romania, reporting severe pain, photophobia, epiphora, and a severe decrease in visual acuity in her right eye. The symptoms started four to five days before the presentation; she linked the debut with eye trauma caused by plant leaves. Visual acuity in the right eye was counting fingers at two meters, IOP was 18 mmHg, and the slit-lamp examination revealed a 6 mm white corneal abscess with perilesional infiltration, thick Descemet folds, Tyndall ++, and myotic pupil. The clinical diagnosis was corneal abscess with exogenous uveitis of the right eye. A SARS-CoV-2 rapid antigen test was performed, and as it was negative the patient was admitted to hospital in the Ophthalmology Clinic.

Conjunctival secretions and corneal scrapings were again collected and sent to the hospital’s laboratory for microbiology diagnosis and antibiogram; standard blood tests were also performed (10.6 × 10^3^/µL leukocytosis and 87.3%/9.2 × 10^3^/µL neutrophilia). Intravenous empirical double antibiotherapy was introduced immediately after sampling (third-generation cephalosporin/beta-lactamase inhibitor—cefoperazone/sulbactam and gentamycin), doubled by local antibiotherapy (topical levofloxacin). Lubricants, mydriatics, topical NSAIDs, and eye patching were also prescribed. The patient presented dizziness, nausea, and a body temperature raised to 37.5 °C; thus, she was booked for examination in the Internal Medicine Clinic. The general examination did not stress any pathological findings, and blood pressure was 100/60 mmHg. Recommendations were as follows: RT PCR SARS-CoV-2 (which was negative), hemoculture if the temperature rises above 38 °C (did not happen), hydration (according to blood pressure), and NSAIDs in case of fever. 

The evolution was unfavorable as the visual acuity decreased to counting fingers at one meter, the corneal abscess remained the same size, a 2 mm hypopyon appeared, and the subjective pain increased (30 April). An infectious disease specialist was asked to see the patient, and the recommendations were as follows: further serological investigations for HIV, cytomegalovirus, and hepatitis viruses B and C, which were negative. It was also recommended to stop cefoperazone/sulbactam + gentamicin therapy and change to a teicoplanin + azithromycin regimen. Despite antibiotic changes, the patient’s state did not improve: visual acuity in the right eye was limited to hand movement, the whole cornea was infiltrated, the abscess was 6–7 mm, and the hypopyon was 3–4 mm (1 May) (Figure 2).

The microbiology results then came in. Bacteriologically, the samples were inoculated under aerobic, anaerobic, and microaerophilic conditions in Columbia blood agar, chocolate agar supplemented with PVX, and MacConkey culture medium for 24 h at 37 °C in aerobiose (48 h in anaerobiose and microaerophily). We observed in both samples on blood agar in large (2–4 mm), convex, non-pigmented, light beige, non-hemolytic, raised aerobiotic colonies, with a dry central portion and wet at the edges (Figure 3).

Biochemically, the strain was glucose-non-fermenting, oxidase-positive, mobile, indole-negative, and urease-positive. The biochemical identification on a VITEK2 automated system (Biomerieux) with a GN card identified *R. radiobacter* with 99% probability after 3.8 h of analysis (Figure 4).

In the microscopic exam we observed long and thin unsporulated Gram-negative rods with a tendency of adhesion between them (Figure 5).

Antibiotic susceptibility testing on the VITEK2 Compact system using the card AST-N233/AST-XN05 showed susceptibility to ticarcillin and ticarcillin–clavulanate (MIC ≤ 8), piperacillin and piperacillin–tazobactam (MIC ≤ 4), cefepime (MIC ≤ 1), imipenem, and meropenem (MIC ≤ 0.25). Aminoglycosides were also susceptible (amikacin MIC ≤ 2, gentamycin MIC ≤ 1, tobramycin ≤ 1), as were the quinolones (ciprofloxacin/ofloxacin MIC ≤ 0.25 and levofloxacin MIC ≤ 0.12). The strain was resistant to colistin (MIC ≥ 16). The MIC for tigecycline was ≤0.5, which the system interpreted as resistant.

On 2 May 2022, the patient requested transfer to the Emergency Eye Diseases Hospital in Bucharest, Romania, and it was granted. The right eye status was as follows: visual acuity was light perception, severe conjunctival hyperemia, infiltrated cornea, 6–7 mm corneal abscess, 4 mm hypopyon, and fundus examination was not possible due to corneal opacity. The clinical and microbiological diagnosis was severe bacterial keratitis (corneal abscess with *R. radiobacter*) and exogenous endophthalmitis. 

Unfortunately for this patient, the evolution was unfavorable, and within days after she was transferred to Bucharest the eye was eviscerated to prevent orbital, sinus, and cerebral infection complications.

## 3. Systematic Review

We searched ProQuest, PubMed, and ScienceDirect for the terms Agrobacterium, Rhizobium, radiobacter, and eye (the exact search string was “(Agrobacterium OR Rhizobium) AND radiobacter and eye”). The eligibility criteria included human studies of infections of the eye with *R. radiobacter*, and all article types except for comments, editorials, etc. The database search yielded 138 studies: 96 from the ProQuest database, 8 from PubMed, and 34 from ScienceDirect; 110 studies were not retrieved as they were non-human studies. We excluded two more studies as they were editorials or comments. Therefore, we selected 26 studies for retrieval: 7 studies from the PubMed database, 10 from ProQuest, and 9 from ScienceDirect; 5 studies were duplicates. We excluded 4 more studies from the analysis after reading the abstracts: two studies identified *R. radiobacter* in solutions used for the storage of contact lenses [29,30]; one study referred to sepsis with *R. radiobacter* originating from a central venous catheter [31]; and one study identified this pathogen in surgical equipment used in ophthalmology (vacuum control manifold) [32] (Figure 6). 

Therefore, we included 17 studies in the systematic review (Table 2).

Rohowetz et al. [33] reported endophthalmitis with *R. radiobacter* in a 79–year-old male patient with type II diabetes mellitus and diabetic retinopathy, who received treatment with intravitreal aflibercept. The patient was treated empirically with intravitreal injections of ceftazidime and vancomycin. After the laboratory results came in, the patient’s therapy was changed to oral azithromycin and levofloxacin. The infection was resolved at 1-month follow-up.

The same author, one year later, reported a case of endophthalmitis in an 85-year-old male associated with insertion of an inferonasal Baerveldt tube [34]. *R. radiobacter* was identified in aqueous humor culture, and was resistant to cefazolin, ceftazidime, amikacin, tobramycin, and trimethoprim–sulfamethoxazole. Because the explantation of the drainage implant was not efficient, pars plana vitrectomy was performed with removal of the intraocular lens, associated with silicone oil infusions and intravitreal antibiotics. The patient’s visual acuity improved after 2 weeks, but then it was lost to follow-up.

Barker et al. [35] reported a series of four cases with *R. radiobacter* in young patients: one after corneal traumatism and three in long-term contact lens wearers. The patients were treated with steroidal anti-inflammatory drugs and with antibiotic drops. All cases had a favorable evolution, with total sight recovery. Furthermore, Fenner et al. [36] reported keratitis in a young patient who was a contact lens wearer.

Another case of *R. radiobacter* infection was reported after phakic intraocular lens implantation in the posterior chamber of the eye in a young patient (29 years old) with myopia [37]. The endophthalmitis was resolved after intravitreal injections of vancomycin and ceftazidime. 

One study identified one strain of *R. radiobacter* (1.14%) in 88 patients expressing symptoms consistent with conjunctivitis [40]. Another study included 44 patients with polymicrobial keratitis, from which *R. radiobacter* was isolated in one case (2.27%).

Nine studies were research articles, which included 912 patients who had undergone cataract surgery [4,5,6,7,8,9,10,11,41]. From those, in 18 cases (1.97%), *R. radiobacter* was isolated. In 7 of those 18 cases, *R. radiobacter* was associated with endophthalmitis, but in the remaining 11 cases it was considered an environmental contaminant. Moreover, Namdari et al. [42] in 2003 reported a case of chronic endophthalmitis after cataract extraction produced by *R. radiobacter*, which was resistant to vancomycin, amikacin, and ceftazidime—drugs often used in empirical therapy. Ultimately, the infection was resolved after changing the therapy to intravitreal injections of gentamycin and oral ciprofloxacin.

Therefore, endophthalmitis with *R. radiobacter* is rare, and is usually encountered after cataract surgery—mostly in old people and immunosuppressed patients, but also in young patients who wear contact lenses for extended periods of time.

## 4. Discussion

*Rhizobium* is a rare agent of endophthalmitis in patients who come into contact with plants [21]. Initially classified as *Agrobacterium*, after the introduction of 16s RNA genetic sequencing in the 1990s, many species were reclassified into the genus *Rhizobium*. The genus is divided into species based largely on pathogenic properties in plants: *R. radiobacter* (non-pathogenic), *R. tumefaciens* (the causative agent of crown gall tumors), *R. rhizogenes* (the causative agent of hairy root disease), and *R. vitis* (the causative agent of tumors and necrotic disease on grapevines). There are also less well-studied proposed species such as *R. rubi* isolated from cane galls on *Rubus* plant species.

The first *Rhizobium* strains were isolated from human infections in 1967 [43]. The first infection with *R. radiobacter* was reported in a case of endocarditis on a prosthetic valve [44]. In 1996, the first case of *Rhizobium radiobacter* endophthalmitis was reported after cataract surgery [42] and intravitreous injections [45]. Few cases have been reported since then [33]. In our clinic, this is the first confirmed case of *R. radiobacter* endophthalmitis. 

The reported cases of ocular *R. radiobacter* infections occurred after traumatic medical procedures involving the eye, such as cataract surgery [42] or intravitreal injections [33]. Our patient, however, developed endophthalmitis after eye trauma caused by a plant leaf. In this case, our understanding is that the first episode was a local infection, which indeed resolved, but acted as a risk factor for infection with *R. radiobacter* after the eye trauma caused by the plant leaf. Indeed, as *Rhizobium* species are predominantly found in soil and plants, it is thought that cases of endophthalmitis after eye surgery are associated with outdoor activities and contamination of the eye with dust. 

Although *R. radiobacter* infections are usually reported in immunocompromised hosts, our patient was immunocompetent; in this case, the eye trauma directly inoculated the eye with the pathogen at a much higher dose than that found in dust that contaminated the eye after medical procedures in other reported cases.

## 5. Conclusions

To summarize, *R. radiobacter* is a soil bacterium that rarely causes human infection. Nevertheless, patients with eye surgery or other procedures that provide a continuity solution of the cornea, including endophthalmitis, can develop endophthalmitis following exposure to soil. It is therefore recommended that patients who undergo surgical procedures such as cataract surgery or intravitreal injections should avoid outdoor activities that involve exposure to soil and plants (yardwork, farming, etc.).

## Figures and Tables

**Figure 1 antibiotics-11-00905-f001:**
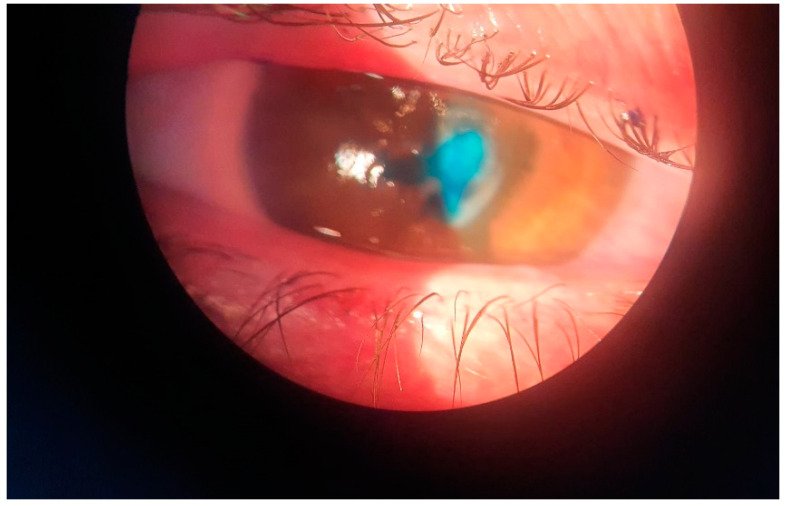
Slit-lamp examination of the right eye showing conjunctival hyperemia, a 4 mm white–yellow central corneal ulceration with infiltrated margins that retains the stain, Descemet folds at the lesion, and endothelial edema (methylene blue staining).

**Figure 2 antibiotics-11-00905-f002:**
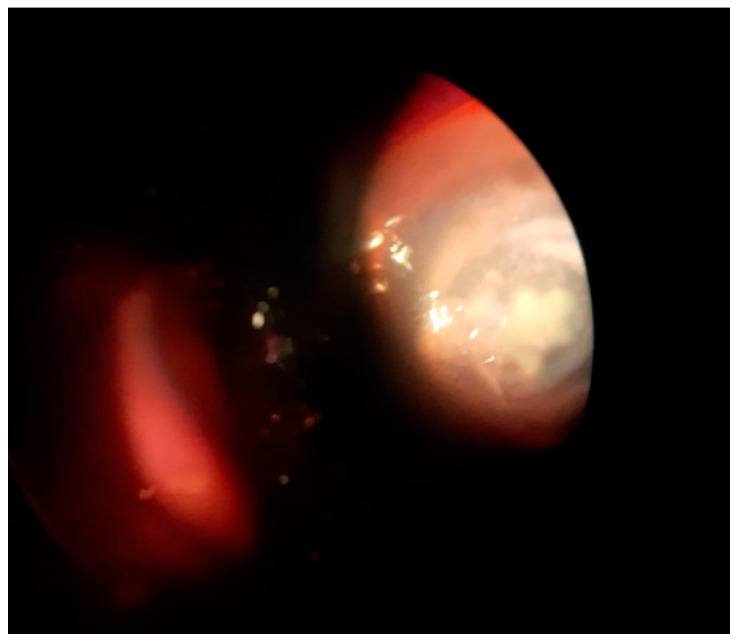
Slit-lamp examination of the right eye highlighting severe conjunctival hyperemia, infiltrated cornea, 6–7 mm central corneal abscess, and hypopyon.

**Figure 3 antibiotics-11-00905-f003:**
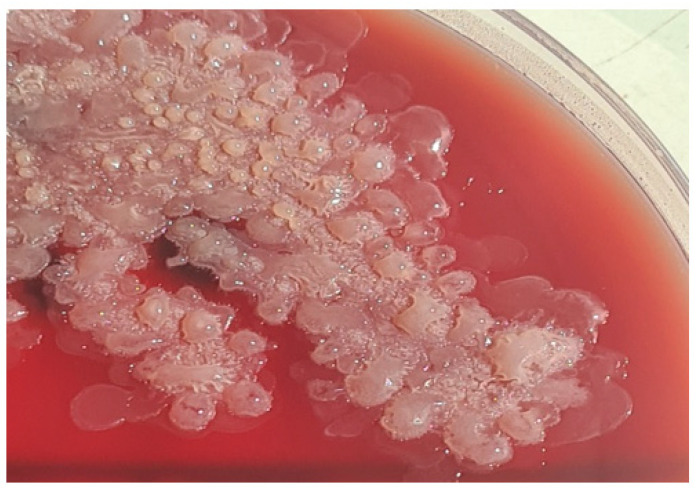
*Radiobacter* colonies on Columbia blood agar.

**Figure 4 antibiotics-11-00905-f004:**
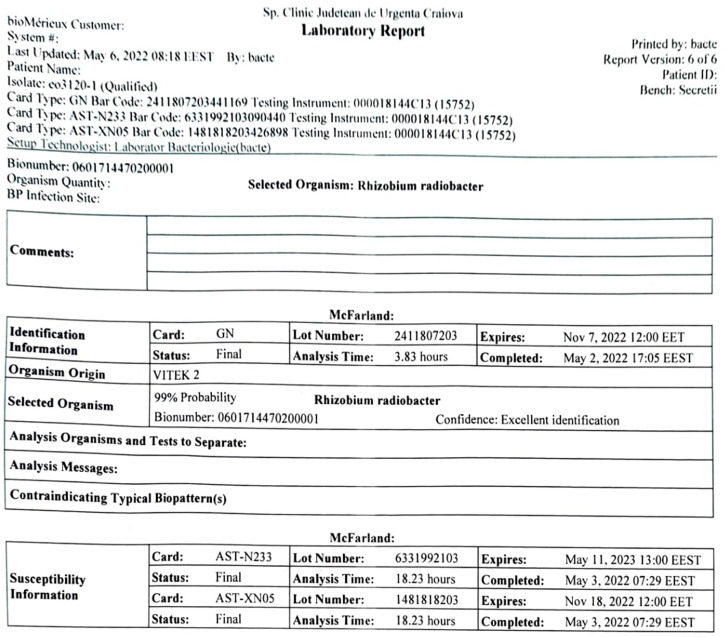
Automated identification results of *Rhizobium radiobacter* on the VITEK2 compact system.

**Figure 5 antibiotics-11-00905-f005:**
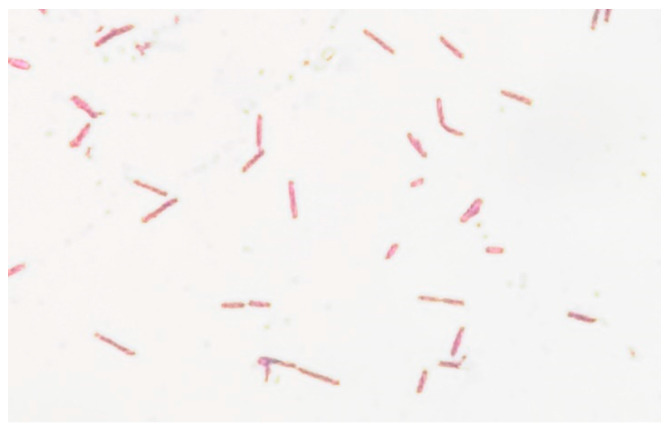
Microscopy of *Radiobacter* culture. Gram stain, magnification 1000×.

**Figure 6 antibiotics-11-00905-f006:**
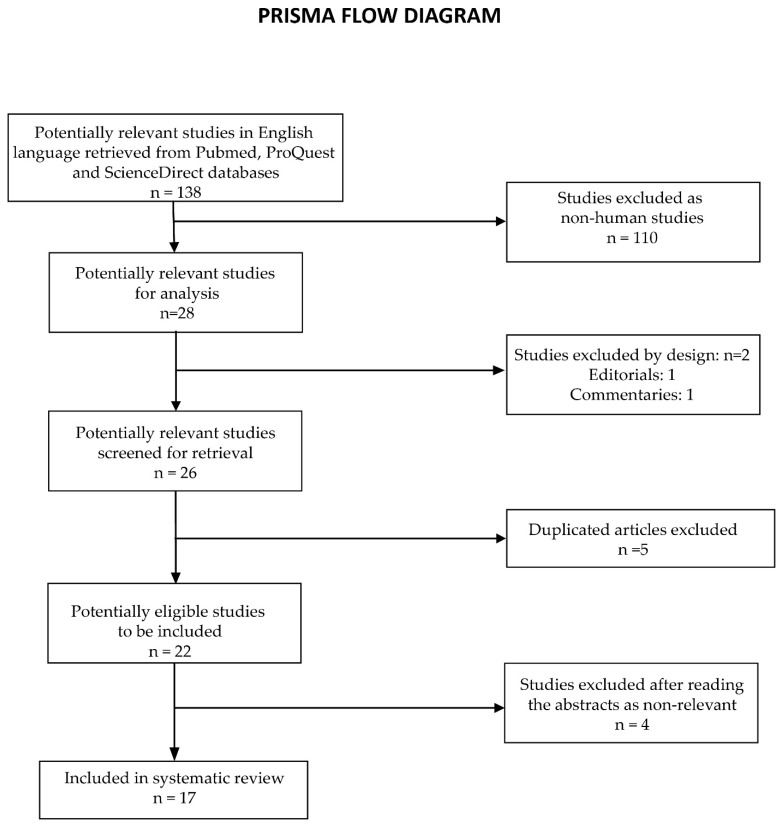
PRISMA flow diagram of the selection process of studies included in this systematic review.

**Table 1 antibiotics-11-00905-t001:** Etiology of endophthalmitis by type.

Endophthalmitis Type	Etiology	Recommended Intravitreal Treatment	Recommended Systemic Antibiotics
After cataract surgery	*Staphylococcus epidermidis, Staphylococcus warneri,* *Staphylococcus capitis, Staphylococcus haemolyticus, Staphylococcus aureus, Propionibacterium acnes, Streptococcus pneumoniae, Streptococcus agalactiae, Streptococcus intermedius, Abiotrophia defectiva, Bacillus cereus, Pseudomonas aeruginosa, Pseudomonas stutzeri, Klebsiella pneumoniae, Haemophilus influenzae, Acinetobacter baumannii, Agrobacterium tumefaciens,* *Citrobacter freundii, Rhizobium radiobacter,* *Escherichia coli, Enterococcus faecalis,* *Stenotrophomonas maltophilia, Mycobacterium chelonae, Propionibacterium acnes* (in chronic cases), fungi *(Candida albicans, Candida parapsilosis, Acremonium strictum, Aspergillus fumigatus, Paecilomyces variotti, Fusarium proliferatum)* [5,6,7,8,9,10,11,12]	Vancomycin, ceftazidime	Rarely used (in severe cases)
After intravitreal injections	*Staphylococcus aureus, Streptococcus viridans, Streptococcus mitis, Streptococcus viridans, Enterococcus faecalis* [3,13]	Vancomycin, ceftazidime	Quinolones (moxifloxacin)
After eye traumatism	*Staphylococcus aureus, S. epidermidis, S. capitis, Staphylococcus warneri, S. pasteuri, S. auricularis, S. piscifermentans, S. lugdunensis, Streptococcus mutans, S. salivarium, S. mutans, S. oralis, S. sanguinis, S. vestibularis, S. termophilus, Enterococcus faecalis, Rhizobium radiobacter, Bacillus cereus, Escherichia coli,* *Candida parapsilosis, Candida albicans, Candida glabrata, Candida famata, Aspergillus flavus, Acremonium curvulum, Fusarium solani, Fusarium proliferatum)* [10,14,15]	Vancomycin, ceftazidime, amphotericin (if suspicion of fungal infection)	Vancomycin, ceftazidime, ciprofloxacin
After Baerveldt tube exposure (in glaucoma treatment)	*Haemophilus influenzae, Streptococcus pneumoniae, Streptococcus agalactia, Streptococcus mitis, Corynebacterium spp., Pseudomonas aeruginosa, Aspergillus flavus, Aspergillus fumigatus* [16]	Aancomycin, ceftazidime	Rarely used
After long-term wear of contact lenses	*Pseudomonas aeruginosa, Serratia marcescens, Staphylococcus epidermidis, Staphylococcus aureus, Streptococcus mitis, Streptococcus salivarium, Streptococcus mutans, Streptococcus pneumoniae, Klebsiella oxytoca, Enterobacter cloacae, Propionibacterium acnes* [17]	Aminoglycosides, fluoroquinolones, 3rd-generation cephalosporins, vancomycin	Rarely used
Endogenous endophthalmitis	*Staphylococcus aureus, Staphylococcus epidermidis, Staphylococcus lugdunensis, Streptococcus viridans, Streptococcus pyogenes, Streptococcus agalactiae, Streptococcus pneumoniae, Enterococcus faecalis, Escherichia coli, Klebsiella pneumoniae, Serratia marcescens, Kingella kingae, Pseudomonas aeruginosa, (Candida albicans, Candida parapsilosis, Candida glabrata, Aspergillus flavus, Aspergillus fumigatus)* [18,19,20]	Vancomycin, ceftazidime (or amikacin)	Antibiotics according to etiology

**Table 2 antibiotics-11-00905-t002:** The studies included in this review.

Authors	Journal	Title	Article Type	No. of Samples/Patients	Mean Age (Years)	Pre-Existent Eye Conditions	Eye Trauma	Antibiotic Susceptibility Results	Drug Therapy	Outcome
Rohowetz et al. (2020) [33]	*Case Reports in Ophthalmology*	Endophthalmitis Caused by *Agrobacterium radiobacter* Following Intravitreal Aflibercept for Diabetic Retinopathy	Case report	1	79	Diabetic retinopathy	Intravitreal aflibercept	-	Intravitreal vancomycin and azithromycin; oral gentamicin and levofloxacin	Infection resolved
Rohowetz et al. (2021) [34]	*Case Reports in Ophthalmology*	*Agrobacterium radiobacter* Endophthalmitis Associated with Baerveldt Tube Exposure	Case report	1	85	Angle-closure glaucoma	Baerveldt tube insertion	Resistance to cefazolin, ceftazidime, amikacin, tobramycin, and trimethoprim–sulfamethoxazole	Injection of intravitreal vancomycin (1 mg) and ceftazidime (2.25 mg). Drops of 0.5% moxifloxacin every 2 h. Injection of 0.2 mg of intravitreal gentamicin	Visual acuity improvement after 2 weeks and then lost to follow-up
Barker et al. (2016) [35]	*Cornea*	*Rhizobium radiobacter*: A Recently Recognized Cause of Bacterial Keratitis	Case series	4	26	None identified	Cement-splash injury	Susceptibility to ciprofloxacin and minocycline	Moxifloxacin drops. Oral ketoconazole. Ciprofloxacin drops hourly and oral ciprofloxacin (250 mg) twice daily	Infection resolved
					26	Contact lens wearer	-	Ciprofloxacin, ceftazidime, levofloxacin, minocycline. Resistant to trimethoprim–sulfamethoxazole	Drops with vancomycin and tobramycin. Vancomycin dropped and ciprofloxacin added	Infection resolved
					19	Contact lens wearer	Corneal ulcer	Susceptibility to gentamicin, ciprofloxacin, levofloxacin, ceftazidime, and polymyxin	Ofloxacin, tobramycin and cefazolin. Cyclopentolate 3 times daily. Prednisolone 1%	Infection resolved with a residual scar
					19	Contact lens wearer	-	Susceptibility to gentamicin, ceftazidime, amikacin, and polymyxin	Oral acyclovir 400 mg and gatifloxacin twice daily. Scopolamine 3 times daily	Lost to follow-up
Fenner et al. (2019) [36]	*American Journal of Ophthalmology Case Reports*	Case of Isolated *Rhizobium radiobacter* Contact-Lens-Related Infectious Keratitis: A Plant Microbe Now Emerging as a Human Pathogen	Case report	1	26	Contact lens wearer	-	Susceptibility to cefepime, ciprofloxacin, and gentamicin	Hourly cefazolin (50 mg/mL) and gentamicin (14 mg/mL). Hourly 0.02% chlorhexidine. Hourly 1.5% levofloxacin, hourly cefazolin, and two-hourly chlorhexidine to both eyes during waking hours, and 0.3% ciprofloxacin ointment overnight. Levofloxacin 1.5% monotherapy for both eyes at four weeks	Favorable evolution with central anterior-to-mid-stromal scar
Al-Abdullah et al. (2015) [37] (no full text available)	*Journal of Refractive Surgery*	Endophthalmitis Caused by *Rhizobium radiobacter* After Posterior Chamber Phakic Intraocular Lens Implantation to Correct Myopia	Case report	1	29	Myopia	Posterior chamber phakic intraocular lens implantation	-	Intravitreal injections of vancomycin and ceftazidime	Infection resolved
Mishra et al. (2019) [4]	*The British Journal of Ophthalmology*	Utility of Broad-Range 16S rRNA PCR Assay Versus Conventional Methods for Laboratory Diagnosis of Bacterial Endophthalmitis in a Tertiary Care Hospital	Research article	8 out of 195 vitreous aspirates from endophthalmitis patients	-	-	Cataract surgery	Susceptible to all tested antibiotics (according to document M45 of CLSI 2010 [38])	-	-
Shirodokar et al. (2012) [6]	*American Journal of Ophthalmology*	Delayed- Versus Acute-Onset Endophthalmitis after Cataract Surgery	Research article	1 out of 119 patients with endophthalmitis	-	-	Cataract surgery	-	-	-
Hsu et al. (2013) [5]	*American Journal of Ophthalmology*	Ocular Flora and their Antibiotic Resistance Patterns in the Midwest: A Prospective Study of Patients Undergoing Cataract Surgery	Research article	1 out of 183 conjunctival cultures	-	-	Cataract surgery	Susceptible to cefazolin, ceftazidime, gentamycin, tobramycin, amikacin, ciprofloxacin, and levofloxacin	-	-
Harbiyeli et al. (2021) [39]	*International Ophthalmology*	Clinical Aspects and Prognosis of Polymicrobial Keratitis Caused by Different Microbial Combinations: A Retrospective Comparative Case Study	Research article	1 out of 44 corneal scrapings	-	Polymicrobial keratitis	-	Susceptible to ciprofloxacin and moxifloxacin	-	-
Haapala et al. (2005) [7]	*Graefe’s Archive for Clinical and Experimental Ophthalmology*	Endophthalmitis Following Cataract Surgery in Southwest Finland from 1987 to 2000	Research article	1 out of 47 patients	-	-	Postoperative endophthalmitis after cataract surgery	-	-	-
Tellegen et al. (2009) [40]	*Journal of Clinical Pathology*	Diagnosis of Conjunctivitis in Primary Care: Comparison of Two Different Culture Procedures	Research article	1 out of 88 patients	-	Infectious conjunctivitis	-	-	-	-
Chiquet et al. (2016) [10]	*The British Journal of Ophthalmology*	Occurrence and Risk Factors for Retinal Detachment after Pars Plana Vitrectomy in Acute Post-Cataract Bacterial Endophthalmitis	Research article	1 out of 123 vitreous aspirates	-	-	Post-cataract bacterial endophthalmitis treated with pars plana vitrectomy	-	-	-
Ambiya et al. (2016) [8]	*Journal of Ophthalmic Inflammation and Infection*	Comparison of Clinico-Microbiological Profile and Treatment Outcome of In-House and Referred Post-Cataract-Surgery Endophthalmitis in a Tertiary Care Center in South India	Research article	1 out of 100 patients			Cataract surgery	Susceptible to amikacin, ceftazidime, gatifloxacin, moxifloxacin, ciprofloxacin, and ofloxacin. Resistant to chloramphenicol	Intraocular antibiotics (1 mg of vancomycin in 0.1 mL of normal saline; 2.25 mg of ceftazidime in 0.1 mL of normal saline). Intravitreal dexamethasone (0.4 mg in 0.1 mL). Topical 1% prednisolone acetate (every 4 h). Cycloplegia. Systemic ciprofloxacin (1500 mg/day in two divided doses)	-
Ness et al. (2011) [9]	*The Journal of Hospital Infection*	Postoperative Nosocomial Endophthalmitis: Is Perioperative Antibiotic Prophylaxis Advisable? A Single Centre’s Experience	Research article	1 out of 16 patients with endophthalmitis	-	-	Cataract surgery	Susceptibility to gentamicin, ciprofloxacin, and ofloxacin	-	-
Chiquet et al. (2009) [41]	*Ophthalmology*	Analysis of Diluted Vitreous Samples from Vitrectomy is Useful in Eyes with Severe Acute Postoperative Endophthalmitis	Research article	1 out of 34 patients with endophthalmitis	-	Diabetes mellitus immunosuppression	Cataract surgery	-	-	-
Friling (2019) [11]	*The Journal of Hospital Infections*	Bacteriology and Cefuroxime Resistance in Endophthalmitis Following Cataract Surgery before and after the Introduction of Prophylactic Intracameral Cefuroxime: A Retrospective Single-Centre Study	Research article	3 out of 95 patients with endophthalmitis	-	-	Cataract surgery	-	Intravitreal injection of 1 mg of cefuroxime; 2.27 mg of ceftazidime and 1 mg of vancomycin. Oral dose of prednisone (60 mg for five days)	-
Namdari et al. (2003) [42]	*Journal of Clinical Microbiology*	*Rhizobium* (*Agrobacterium*) *radiobacter* Identified as a Cause of Chronic Endophthalmitis Subsequent to Cataract Extraction	Case report	1	62	Uncomplicated cataract extraction	-	Susceptibility to ciprofloxacin, resistance to ceftazidime and vancomycin	Intravitreal injection of amikacin (0.4 mg) and vancomycin (1 mg). Intravitreal administration of gentamicin (0.4 mg). Oral ciprofloxacin (500 mg twice daily for 10 days)	Infection resolved

## Data Availability

The data presented in this study are available upon request from the corresponding author.

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
