# Peer review of "A Rare Case of Endophthalmitis with Rhizobium radiobacter, Soon after a Resolved Keratitis: Case Report"

_antibiotics, 2022, doi:10.3390/antibiotics11070905_

Round 1

Reviewer 1 Report

This review report paper by Andrei et al is written well and the authors have done extensive literature search. This paper has significant merit for publication. My suggestion is if author add a table showing similar kind of complication caused by other bacteria’s. This will highlight the overall importance of Rhizobium Radiobacter in establishing this disease. It will also help readers who are studying Endophtalmitis.

Reviewer 2 Report

Title: change endophtalmitis to endophthalmitis

line 33: change endophtalmitis to endophthalmitis

line 35: change endophtalmitis to endophthalmitis

line 39: Gram in "gram-negative" should be capitalized as it is someone's last name

line 43: "most in immunocompromised patients" can be changed to "most often in immunocompromised patients"

line 43: "It were reported a few cases of..." What was reported? Sentence doesn't make sense

line 53: change "immunocompromising condition" to "immunocompromised condition" 

line 53-55: what is normal or good visual acuity? how does this measurement compare to the US measurement of 20/20? What is normal IOP? Is this high or low or normal?

line 112: Change "Infectious Disease Doctor" to "Infectious Disease specialist" or "doctor". Doctor does not need to be capitalized

line 166: Italicize R. radiobacter

line 167: Italicize R. radiobacter

line 178: change endophtalmitis to endophthalmitis

line 185: change endophtalmitis to endophthalmitis

line 193: change "corneal traumatism" to "corneal trauma"

line 199: change endophtalmitis to endophthalmitis

line 206: change endophtalmitis to endophthalmitis

line 212: change endophtalmitis to endophthalmitis

line 217: change endophtalmitis to endophthalmitis

line 227: change endophtalmitis to endophthalmitis; Italicize Rhizobium radiobacter

line 236: change endophtalmitis to endophthalmitis

line 245: change "corneal traumatism" to "corneal trauma";  change endophtalmitis to endophthalmitis

line 246-247: remove "the" before cataract surgery 
